# The influence of sexual arousal on subjective pain intensity during a cold pressor test in women

Lara Lakhsassi[1]*, Charmaine Borg[1], Sophie Martusewicz[1], Karen van der Ploeg[2], Peter J. de Jong[1]

1 Department of Clinical Psychology and Experimental Psychopathology, University of Groningen, Groningen, The Netherlands, 2 University Medical Center Groningen, Groningen, The Netherlands

* l.lakhsassi@rug.nl

**Data Availability Statement:** Data available in the DataverseNL repository. DOI: https://doi.org/10.34894/BRZT3O.

## Abstract

### Background & objectives

Pain can be significantly lessened by sex/orgasm, likely due to the release of endorphins during sex, considered potent analgesics. The evidence suggests that endorphins are also present during sexual arousal (that is, prior to sex/orgasm). It follows then that pain can be modulated during sexual arousal, independent of sex/orgasm, too. Accordingly, sexual arousal induced by erotic slides has been demonstrated to lessen pain in men, but not in women. One explanation could be that for women, the erotic slides were not potent enough to elicit a lasting primed state of sexual arousal by the time pain was induced. Thus, the current study aims to optimize the means of inducing a potent state of sexual arousal and subsequently examine the potentially analgesic influence of sexual arousal on pain in women. As a subsidiary aim, the study also assesses whether the anticipated analgesic effect of sexual arousal would be stronger than that of distraction or generalized (non-sexual) arousal.

### Methods

Female participants ($N$ = 151) were randomly distributed across four conditions: sexual arousal, generalized arousal, distraction, neutral. Mild pain was induced using a cold pressor while participants were concurrently exposed to film stimuli (pornographic, exciting, distracting, neutral) to induce the targeted emotional states. A visual analogue scale was utilized to measure the subjective level of pain perceived by the participants.

### Results

Sexual arousal did not reduce subjective pain. Generalized arousal and distraction did not result in stronger analgesic effects than the neutral condition.

### Conclusion

The present findings do not support the hypothesis that sexual arousal alone modulates subjective pain in women. This might be due to the possibility that genital stimulation and/or

**Funding:** The author(s) received no specific funding for this work.

**Competing interests:** The authors have declared that no competing interests exist.

orgasm are key in pain reduction, or, that feelings of disgust may inadvertently have been induced by the pornographic stimulus and interfered with sexual arousal in influencing pain.

## Introduction

"Pleasure is an entity that antagonizes pain"

– Komisaruk & Whipple (1986).

There is considerable evidence that sexual stimulation can reduce the response to painful stimuli. Animal studies have shown that pain-related reflexes and vocalizations in response to noxious stimuli (e.g., tail/skin shock, pinch stimulation) were suppressed in female rats when vaginal cervix probing (CP) was applied [1–3]. The apparent CP-induced analgesia was further found to be more effective than a 2-mg/kg dose of morphine sulphate at suppressing responses to painful foot compression on a yeast-infected paw [4]. These findings were paralleled by a study on male rats, depicting an analgesic effect to shock and pinch during copulation, also measured by reduced pain-related vocalizations [5]. In humans, these effects were studied in women in the collaborative work of Whipple and Komisaruk, who demonstrated that the thresholds for pain tolerance and pain detection–as measured by a calibrated finger-compressing device increasing gradually in pressure–were significantly increased when paired with genital self-stimulation, and considerably more so when achieving orgasm [6]. Most notably, they had subsequently found similar effects without the use of self-stimulation, provided that orgasm could be reached by means of self-induced imagery [7]. This led them to the conclusion that sexual pleasure is a powerful modulator for pain [8].

The observed analgesic effect that sex has on pain has been linked to the release of opioids and endorphins due to their potent analgesic effects [5, 6, 9]. Whilst in the sexual context, opioid and endorphin release has most often been documented during sexual activity and orgasm, the release of opioids and endorphins may indeed be present during sexual arousal before any sexual activity takes place. For instance, after an infusion of beta endorphins, male rats are increasingly motivated to chase after a sexually receptive female rat, contrary to rats who were administered saline or naloxone, an opioid receptor antagonist [10]. It may be inferred that the rats infused with endorphins were sexually aroused, driving them to pursue the female. In the same vein, subjective sexual arousal in humans has been found to precipitate somatic motor system responses that in turn prepare a person for action [11]. Taken together, it seems plausible to predict that sexual arousal alone (i.e., without direct genital stimulation, and in the absence of orgasm) might also modulate pain.

Yet, recent studies examining whether similar pain reducing effects are also evident when sexual arousal is elicited (i.e., without direct genital stimulation, and in the absence of orgasm) have shown paradoxical findings. On the one hand, some of the evidence shows that upon being primed with an erotic auditory stimulus, pain thresholds on a cold pressor test (CPT) are lowered (rather than increased) in women while they had no influence in men [12]. On the other hand, and, in support of the view that sexual arousal can reduce pain in the absence of direct stimulation or orgasm, it was found that upon being primed by erotic photographs, self-reported pain thresholds during a subsequent CPT were increased. However, this analgesic effect was restricted to the male participants and was not found for women [13]. One potential explanation for the mixed findings might be that the level of sexual arousal elicited by the auditory and pictorial stimuli used in these earlier studies was not sufficiently high to systematically

affect participants' pain responses. In line with this, a meta-analysis showed that still erotic pictures or erotic audiotaped stimuli yielded weaker arousal responses in women than pornographic films. It also provides evidence that erotic photographs are sufficient for inducing subjective sexual arousal in heterosexual men, but not in heterosexual women [14], which might explain why the pictorial stimuli had successfully reduced pain in men but not in women [13]. Thus, in order to assess whether a potential modulatory effect of subjective sexual arousal on pain is also possible in women, the current study focused on female participants and accordingly utilized a pornographic video highly appraised by heterosexual women in order to ensure a more potent level of sexual arousal. Furthermore, in order to ensure that the intended motivational state would persist during the pain-eliciting task, the pornographic video was not only presented prior to the pain stimulation, but also continued after starting the CPT that was used to elicit pain.

As a subsidiary aim, the present study also examined the specificity of the impact of sexual arousal on pain by discerning it from potentially distracting and generally (non-sexually) arousing properties, which may also influence pain [15, 16]. To compare the impact of sexual arousal on pain with that of general distraction on pain, we added a control condition in which a neutral (non-emotional) film was paired with a mildly taxing cognitive task to infer a state of non-sexual distraction. Next, we compared the influence of sexual arousal with the impact of generalized (i.e., non-sexual) arousal and a neutral taskless control condition. Thus, the present study involved three comparator stimuli/conditions: distraction, generalized arousal, and a neutral reference (baseline) condition.

We hypothesized that sexual arousal may be a key pain modulator, regardless of genital stimulation or orgasm. We additionally hypothesized that sexual arousal, in comparison to mere distraction and generalized arousal, would result in relatively low subjective pain during a CPT. In short, we hypothesized that: (i) participants in the sexual arousal condition would report less pain during the CPT than participants in the neutral condition, (ii) participants in the sexual arousal condition would also report less pain than those in the mere distraction and general arousal conditions, and (iii) that, in line with previous research, participants in the distraction and general arousal conditions would report less pain than those in the neutral control condition [16–19].

## Methods

### Participants

Female participants (*N* = 164) between were randomly recruited from the University of Groningen participant pool for first-year psychology students (*n* = 60), as well as by means of social media advertisements in order to include non-students (*n* = 104). Recruitment criteria involved predominantly heterosexual (i.e., self reported being 50% or over on a scale from completely homosexual to completely heterosexual) and English-speaking women.

Participants excluded from statistical analyses involved students who had previously participated in the pilot study (*n* = 2), participants whose participation failed to meet the standardized protocol due to researcher or technical error (*n* = 6), and participants who were not predominantly heterosexual, or self-reported a rating of being 49% and under in a scale from completely homosexual to completely heterosexual (*n* = 5). Thus the final sample that was subjected to the statistical analyses consisted of 151 predominantly heterosexual (*M* = 84.98, *SD* = 13.92) participants between ages 18 and 29 (*M* = 20.81, *SD* = 2.10). The initial sample size intended in order to reach a statistical power of .80 to detect a difference between conditions with an effect size of 0.25 (i.e., *N* = 180) within a Fixed-Effects One-Way Analysis of Variance (ANOVA) with four conditions, as determined by G*Power 3.1, could unfortunately not be

met due to force majeure (i.e., COVID-19 global pandemic). Based on the actual number of participants we reached a power of .72 to detect the hypothesized difference (with a Cohen's d effect size of 0.5, $\alpha$ = 0.05) between conditions. All participants were given the option to earn either a financial reward or student credits in compensation for their time. The study was approved by the Ethical Committee of Psychology (ECP approval code: PSY-1920-S-0052).

## Materials

**Film stimuli.**   In order to induce the target emotional states for each of the four conditions, we used film stimuli that were administered on a desktop screen (~60x40 centimeters), To elicit *sexual arousal*, we selected a sexually explicit film featuring a heterosexual couple engaging in foreplay (2min30) and intercourse (3min30). To elicit *generalized arousal*, we selected a parkour compilation film stimulus featuring men performing extreme obstacles. To induce a *neutral (reference) state*, a black-and-white film stimulus featuring a train traveling in the outdoors was used. To elicit a state of *distraction* away from pain, we presented the same black-and-white film and combined it with a counting task [20]. The choice of films used for eliciting (sexual) arousal (i.e., porn and parkour films) were selected by the research team upon consensus that these would successfully elicit the target emotional states. The choice of the film used to elicit a neutral or distraction state was selected based on a series of previous studies within our research group in which the train film was successfully used [21]. All films had a (maximum) duration of six minutes.

**Cold pressor test.**   A CPT was constructed in the laboratory in order to simulate the sensation of peripheral pain [22]. The set-up involved a plastic container with a built-in separator that allowed for two compartments (Length: 52 cm, Width: 36 cm, Height: 12 cm) filled approximately halfway with water in order to ensure that the participant's hand could be fully submerged. Medium-sized ice cubes were placed into the container in order to lower the temperature to a standardized level varying between 4 and 4.3 degrees Celsius. An aquarium pump was steadied onto the bottom of the container in order to circulate the water around the separator. Lastly, a thermometer was secured onto the compartment closest to the participant in order to monitor the target temperature. An ice-maker machine and refrigerator were readily available in the laboratory in order to continuously generate ice cubes between participants and store them for the next testing day.

## Measures

**Manipulation check.**   A computerized visual analogue scale (VAS) ranging from 0 (= not at all) to 100 (= very) was employed to assess whether the target emotional state relevant to the film stimulus was successfully induced prior to beginning the CPT. Accordingly, participants in the SEX and NEUTRAL conditions were asked to rate their level of sexual arousal; those in the PARKOUR condition were asked to rate their level of general (non-sexual) excitement, while those in the COUNTING condition were asked to rate their level of cognitive involvement with regard to the counting task in order to subsequently infer distraction from the CPT [20].

**Pain.**   Pain intensity (i.e., subjective pain) experienced at the end of the CPT ("how painful was the water?") was measured with an on-screen VAS ranging from 0 (= 'not at all painful') to 100 (= 'very painful').

## Procedure

In a between-group experimental design, participants were randomly assigned to one of four conditions, namely: SEX (*n* = 38), PARKOUR (*n* = 35), COUNTING (*n* = 35), and NEUTRAL

($n$ = 43). Upon their arrival, participants were invited to read the information and consent forms and signed upon agreement. In order to circumvent the possibility of demand bias, the true purpose of the study was not completely revealed. Instead, the participants were under the impression that the study concerned testing the role of sexual arousal on motivation.

Next, participants were asked to fill in a pre-experiment Qualtrics descriptive questionnaire in order to assess eligibility criteria. This included the following questions: "how old are you?", and "on a scale ranging from 0 to 100, how heterosexual do you consider yourself?", where 0 denoted 'not heterosexual at all' and 100 denoted 'completely heterosexual'. During this time, the researcher was responsible for ensuring that the water temperature in the CPT was at ~3.7 degrees Celsius by the time that the participant had finalized the questionnaire and been invited into the experiment cubicle; this allowed for the temperature to increase to the targeted 4–4.3 degrees by the time the experiment started (that is, it allowed the researcher some time to explain the procedure and answer the participant's questions). In order to accomplish this, the researcher was equipped with: (a) ice cubes to lower the temperature, (b) tepid water to warm it, and (c) a small cup to remove the remaining ice cubes prior to beginning the experiment, as well as to ensure that the water volume was precisely at the marked level. The aquarium pump was to be switched on in order to circulate the water across both compartments during this time. This process was not visible to participants.

Once participants were invited into the experimental cubicle, they were seated facing the desktop, computer mouse, and the CPT on their right-hand side. The film stimulus to which they were assigned was then disclosed (i.e., a pornographic film, a parkour film, a train film with or without a counting task). Participants in the COUNTING condition were instructed to count the number of poles displayed on both sides of the road in the film and remember this number, for they would be asked to report it in the post-experiment questionnaire (this was a prompt to ensure that they kept cognitively busy during the experiment (although the number of poles counted did not really matter, 66% of participants correctly counted 60+ poles, while 22% counted 15–60 poles, and 12% counted less than 15 poles). Next, the researcher provided a detailed explanation to the participant of what was to be expected from the experiment. Here, the participant was informed that they would privately view the film stimulus in the dark room. A little while into viewing the film, a VAS (i.e., manipulation check) would appear at the bottom of the screen, underneath the film. This VAS always appeared after 1 minute and 45 seconds of watching their respective film. Afterwards, a brief instruction appeared at the bottom of the screen requesting that they commence the CPT (i.e., "Place your hand in the water") while continuing to watch the film simultaneously. This instruction always appeared after two minutes of watching their respective film in order to allow the participant enough time to immerse themselves in the film, and for the target emotional state to thereby be induced prior to beginning the CPT. Before the experiment, the participant was informed that the water would be very cold and likely feel uncomfortable, and that the task required them to submerge their entire hand flatly at the bottom of the container, with all fingers spread, for as long as they could tolerate. Once participants could no longer tolerate the cold water and removed their hand from the CPT, the film would immediately stop playing, and participants were to dry their hand using the available paper towel, and respond to the post-experiment VAS (i.e., pain measure) that would immediately appear on the screen. The maximum time that the participants could use the CPT was 4 minutes.

Finally, participants were to follow the on-screen instructions leading them to a post-experiment VAS and questionnaire, which served as an additional manipulation check for the induced emotional state throughout the experimental manipulation. This included the following questions: "how sexually aroused were you during the experiment?"; "how distracting was the video (and counting task, if applicable) from the cold sensation of the water?"; "how excited

(non-sexually) were you during the experiment?"; and "if you were in the counting group, how many poles did you count?". With the exception of the last question, which requested a fill-in-the-blank response, all other questions were to be answered through ratings on a VAS from 0 to 100.

Prior to leaving the participant to begin the experiment, the researcher would provide the participant with the available headphones and instruct them to press the space bar as soon as they were ready to begin (i.e., once the lights were switched off and the researcher had left the laboratory space). Upon the end of the experiment, the program would instruct the participant to exit the private space and inform the researcher that they were finished. See Fig 1 for a visual illustration of the precise timing for each step within the experiment.

E-prime software was utilized throughout the experiment in order to provide the participants with procedural instructions (i.e., displaying VASs and commands relevant to the start and end of the CPT), present the programmed film stimuli corresponding to each condition, as well as to record participant data relevant to task duration.

### Data analyses

**Manipulation checks.** An independent-samples t-test was conducted using the mid-experiment VAS responses of sexual arousal as the dependent variable in the SEX and NEUTRAL conditions to illustrate how participants were feeling after 1 minute and 45 seconds of watching their respective film (and prior to beginning the CPT). Additionally, descriptive statistics were computed to verify the effectiveness of the PARKOUR and COUNTING conditions in eliciting non-sexual excitement and cognitive involvement in the counting task, respectively. As a supplementary manipulation check, we subjected the post-CPT VAS scores of subjective sexual arousal, distraction, and generalized arousal to one-way ANOVAs to verify if the experimental film stimuli had indeed successfully induced the target emotional states relevant to each of the four conditions.

**Hypothesis testing.** Prior to conducting hypotheses testing, ANOVA assumptions were assessed via Q-Q plots and histograms, in order to determine whether the normality of residuals assumption was met, as well as the Levene's test, in order to determine whether the homoscedasticity assumption was met. In order to test the hypothesis that sexual arousal in women decreases pain intensity, we have conducted a between-group one-way ANOVA on subjective pain ratings across conditions. To determine whether the impact of sexual arousal would be stronger than that of distraction and generalized arousal, we tested planned comparisons between the SEX and COUNTING conditions, as well as between the SEX and PARKOUR conditions. To determine the hypothesized pain-reducing effects of distraction and generalized arousal, we tested the planned comparisons between COUNTING and NEUTRAL and between PARKOUR and NEUTRAL, respectively. All statistical analyses were performed on SPSS Statistics (version 28).

## Results

### Manipulation checks

Likely due to their involvement in the presented film, 16 participants failed to respond to the mid-experiment VAS presented at 1.45 min after the start of the film (n = 8 in the SEX, n = 1 in NEUTRAL, n = 2 in PARKOUR, and n = 5 in COUNTING condition) and could therefore not be included in the manipulation check on the basis of the mid-experiment VAS. In support of the effectiveness of the sexual arousal manipulation, an independent-samples t-test on the available mid-experiment VAS ratings of sexual arousal showed that subjective sexual arousal was substantially higher in the SEX condition (M = 61.80, SD = 25.77) than in the NEUTRAL

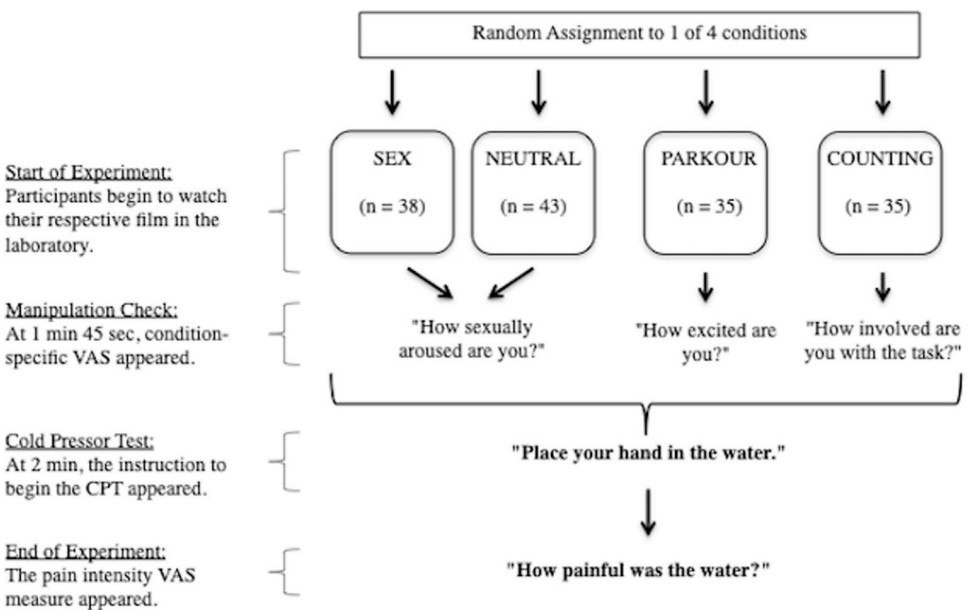

**Fig 1. Flow diagram: Experiment, step-by-step procedure.**

condition ($M = 18.81$, $SD = 21.48$) prior to beginning the CPT ($t$ (55.28) = 7.47, $p < .001$, equal variances not assumed, Cohen's d = 1.84). An inspection of the mean arousal ratings in the PARKOUR condition ($M = 53.03$, $SD = 23.84$) and the cognitive involvement ratings in the COUNTING condition ($M = 77.90$, $SD = 19.54$) confirmed that these comparison conditions were also successful in eliciting the target emotional states prior to beginning the CPT.

The post-experiment VAS rating provided further support for the effectiveness of the experimental conditions to elicit the target emotional states. One-way ANOVAs showed significant differences between conditions for each of the target states (see Figs 2–4): Sexual arousal ($F$ (3, 147) = 65.67, $p < .001$, partial $\eta^2 = 0.57$), generalized arousal ($F$ (3, 147) = 6.91,

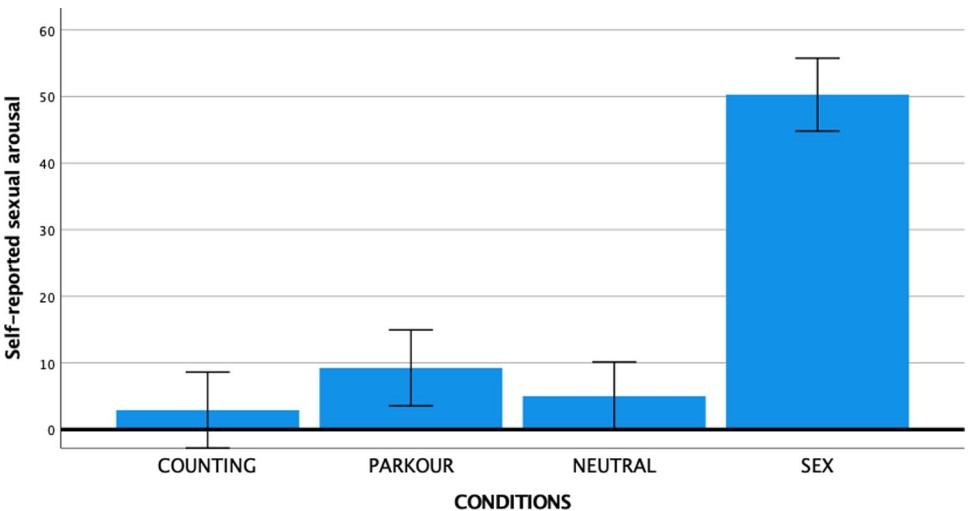

**Fig 2. Mean self-reported sexual arousal levels per condition with error bars representing 95% confidence interval.**

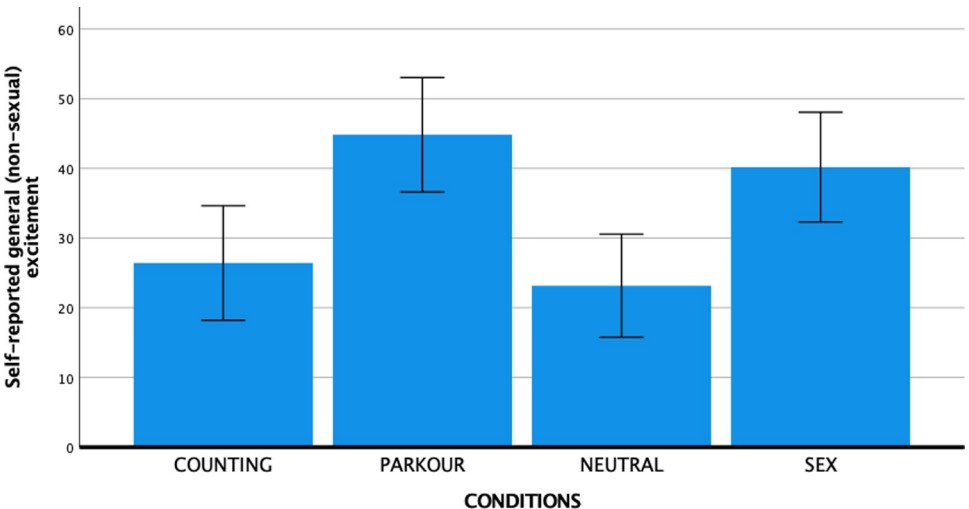

**Fig 3. Mean self-reported generalized arousal per condition with error bars representing 95% confidence interval.**

$p < .001$, partial $\eta^2 = 0.12$), and distraction ($F(3, 147) = 6.28$, $p < .001$, partial $\eta^2 = 0.11$). Supporting the specificity of the sexual arousal eliciting effect of the SEX condition, follow-up analyses with Bonferroni corrections showed that subjective sexual arousal was substantially and statistically significantly higher in the SEX condition ($M = 50.29$, $SD = 26.58$) compared to the PARKOUR condition ($M = 9.26$, $SD = 16.22$, $p < .001$), the COUNTING condition ($M = 2.91$, $SD = 7.94$, $p < .001$), and the NEUTRAL condition ($M = 5.00$, $SD = 11.51$, $p < .001$). Generalized arousal was significantly higher in the PARKOUR condition ($M = 44.83$, $SD = 25.70$) compared to the COUNTING ($M = 26.43$, $SD = 28.40$, $p = .013$) and NEUTRAL ($M = 23.16$, $SD = 20.95$, $p = .001$) conditions, but not compared to the SEX condition ($M = 40.18$, $SD = 23.52$, $p = 1.00$). Distraction was stronger in the COUNTING condition ($M = 40.11$, $SD = 29.51$) than in the NEUTRAL condition ($M = 22.47$, $SD = 22.87$, $p = .030$), but overlapped substantially across the other two conditions such that the COUNTING condition did not differ significantly from the SEX condition ($M = 47.95$, $SD = 31.31$, $p = 1.00$) or the PARKOUR condition ($M = 34.54$, $SD = 24.87$, $p = 1.00$).

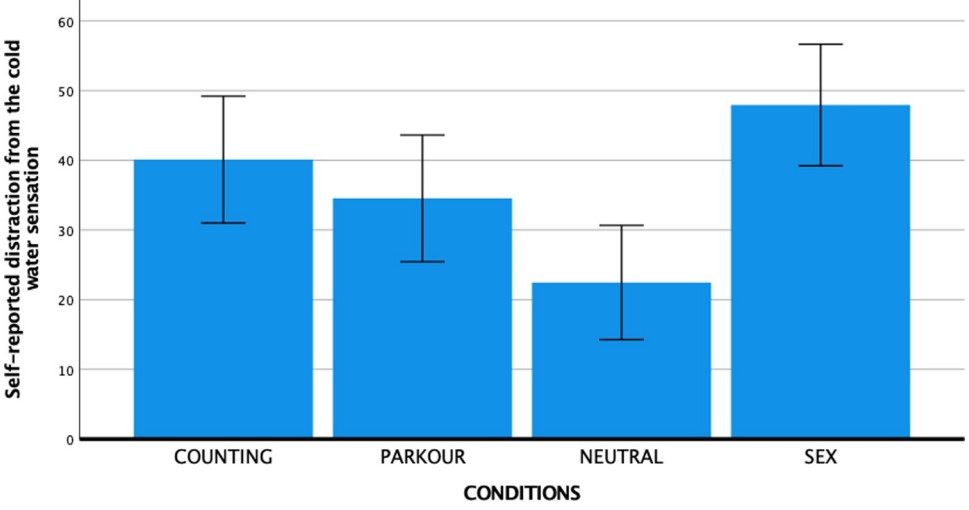

**Fig 4. Mean self-reported distraction levels per condition with error bars representing 95% confidence interval.**

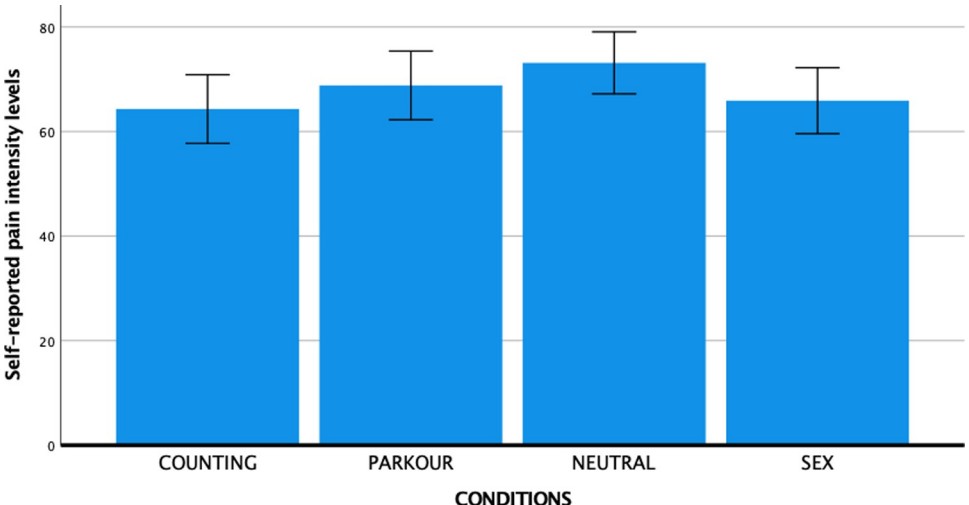

**Fig 5. Graphical illustration of subjective pain ratings per condition with error bars representing 95% confidence interval.**

### Influence of sexual arousal on subjective pain

The analyses of ANOVA assumptions via histograms and Q-Q plot showed that the residuals across conditions for subjective pain were negatively skewed. Yet, given the relatively large sample size (i.e., SEX: $n$ = 38, COUNTING: $n$ = 35, PARKOUR: $n$ = 35, TRAIN: $n$ = 43), an ANOVA remains robust against deviations from normality [23]. Levene's test based on the mean for subjective pain showed that the homogeneity of variances assumption was not violated in our sample (Levene's statistic (3, 147) = 0.69, $p$ = .56).

Mean pain scores per condition are presented in Fig 5. A one-way ANOVA showed that there were no significant differences in subjective pain between conditions ($F$ (3, 147) = 1.54, $p$ = .21, partial $\eta^2$ = 0.03). Although the pattern of findings was in the predicted direction, the differences between the SEX and the comparison conditions were of small to medium effect size and did not reach significance. More specifically, the difference between the SEX and NEUTRAL condition (hypothesis 1) was of small to medium effect size (d = 0.35), the difference between SEX and PARKOUR was very small (d = 0.14), as was the difference between the SEX and COUNTING condition (d = 0.08) (hypothesis 2). The difference between the COUNTING and NEUTRAL condition was of a small to medium effect size (d = 0.47), and the difference between the PARKOUR and NEUTRAL condition was small (d = 0.23) (hypothesis 3).

## Discussion

The core aim of this study was to test whether, in the absence of orgasm or genital stimulation, sexual arousal would reduce pain responses in female participants. The study was designed to test potential explanations for the failure of previous research to find robust analgesic effects of sexual arousal in women. Instead of eliciting sexual arousal via presenting the sex stimulus before the pain induction procedure, we continued to present the sexual arousal-eliciting stimulus during the pain stimulation in order to assure that the targeted motivational state would not (partly) subside during the pain stimulation. In addition, we used an audiovisual stimulus (erotic clip), as this is deemed more potent in eliciting arousal in women than slides or an audiotape [14]. The film clip we used was highly successful in eliciting subjective sexual arousal

both prior to beginning the CPT (M > 60), as well as throughout the experiment (M > 50). The CPT, too, was found to be highly effective in eliciting pain, and the mean pain rating in the NEUTRAL condition (M > 70) provided ample room for being reduced by effective moderators. Yet, despite the erotic clip successfully eliciting substantial subjective sexual arousal, the difference in pain ratings between the SEX and NEUTRAL conditions was not significant and of small to medium effect size. Thus, subjective pain was not systematically reduced in the SEX condition–although the pattern of findings was in the predicted direction.

While the findings in this study corroborate previous research showing that subjective sexual arousal per se is not sufficient to elicit robust analgesic effects in women, it has also been hypothesized that the lack of a pain modulatory effect may be attributable to female participants having experienced feelings of disgust alongside sexual arousal [13]. Indeed, women are substantially more likely to respond to pornography with negative affect such as disgust [13, 14, 24–26]. Germane to this, it has been found in previous research that for women, pornographic stimuli elicited similar brain responses as prototypical disgusting stimuli. Moreover, it has been found that the brain responses to penile-vaginal penetration pictures were modulated by the level of subjective disgust elicited by these sex stimuli. [27, 28]. If indeed such avoidance-related affective states were elicited by the current SEX condition, these might have counteracted the influence of approach-related appetitive affective states (e.g., sexual arousal) resulting in exacerbated pain [29], rather than inhibited pain. In other words, if indeed disgust was elicited along with sexual arousal, this might have counteracted the pain reducing effects of sexual arousal and thus nullified the net effect of the manipulation on pain. Future studies might therefore consider attempting to minimize this risk by critically assessing the pornographic stimuli used for female participants such that they elicit as little aversive affect as possible. Accordingly, studies show that female-centric pornography–typically involving a lengthier duration of foreplay and an emphasis on female pleasure–significantly increases subjective sexual arousal in women in comparison to male-centric "commercial" pornography whilst minimizing aversive affect [14, 25, 26]. Taken together, while the current study adds to the available evidence that subjective sexual arousal per se is not sufficient to elicit robust analgesic effects in women, we cannot exclude the possibility that no effect was found in women due to (concurrent) negative affect associated with the pornographic stimulus. In future studies, an assessment of negative affect during the experiment, alongside an implementation of female-friendly stimuli, is recommended as a potentially helpful means of (statistically) controlling for disgust when attempting to induce sexual arousal in women. In the event that the implementation of these recommendations continue to show no pain modulation in women during sexual arousal, follow-up studies would be necessary to test whether this conclusion still holds with alternative sex stimuli. For instance, earlier studies show that direct tactile stimulation can successfully elicit sexual arousal to initially neutral stimuli [30]. Such an approach may also have the advantage that it minimizes the chances of inadvertently eliciting negative affect like disgust, and may therefore be especially effective in assessing the impact of sexual arousal on pain in female participants.

In the current design we included a COUNTING and PARKOUR condition to examine the specificity of the effects of sexual arousal on participant pain responses. While in the absence of an effect of sexual arousal on pain, these conditions became superfluous as comparison conditions, the impact of distraction and generalized arousal may still be relevant in their own right. The manipulation check data confirmed the efficacy of both the COUNTING and PARKOUR conditions when compared to the NEUTRAL control condition, yet both distraction and generalized arousal did not result in lowered subjective pain. One explanation might be that, similar to the rationale offered for the SEX condition, the PARKOUR stimulus at a rating-level comparable to that of sexual arousal might not be sufficient to have an effect on pain.

As presumed with regard to the influence of sexual arousal on pain, it may be that generalized arousal, too, necessitates a physiological component prior to influencing pain. As for the COUNTING condition, the literature suggests that distraction does not always influence subjective pain. Rather, this may depend on the type of distraction [31], which, in this study, was peripherally distracting (i.e., requiring working memory representations of sensory information), as opposed to centrally distracting (i.e., requiring cognitive processes often abstract in nature) [32]. In turn, because pain processing is a controlled task, controlled central attention is necessary in order to successfully divert attention away from pain and reduce its perceived intensity [33]. Otherwise, a reverse effect may occur in which pain distracts from the cognitive task [34], which could explain why 34% of participants counted less than 60 poles. That said, it is noteworthy that the NEUTRAL condition may have elicited distraction on its' own, irrespective of any counting task. If this is the case, this may have reduced the current study's sensitivity to identify additional moderating effects of distraction from pain via the counting task.

## Limitations

This study had some limitations that should be considered. First, the target sample size was not reached. This implied that the a priori power reduced from .80 to .72, which slightly reduced its sensitivity to reliably identify differences of medium effect size. Furthermore, although the films were selected for efficacy in inducing the target state (i.e., sexual/non-sexual arousal, distraction), it cannot be ruled out that other emotions/responses, which were not controlled for, might have been elicited as well, thereby possibly counter-forcing the effects of targeted states. Next, due to technical problems, the study does not have data that can be analyzed with regard to the time that participants kept their hand in the CPT (thus, pain tolerance was not measured). It can therefore not be ruled out on the basis of the available data that the pain reducing effect of sexual arousal (or distraction) might have increased the duration that they could tolerate the cold water while maintaining a similar level of pain at the time of withdrawal. Future research including both a measure of tolerance and pain at withdrawal is necessary to test the validity of this speculation. Finally, although a fixed water temperature was used across experimental conditions, and random allocation to experimental conditions was utilized in order to prevent an unequal spread of individual differences in pain thresholds between conditions, it cannot be ruled out that perhaps some a priori differences between conditions might have been present, in turn reducing the sensitivity of the design to find differences in subjective pain between conditions. Future studies may consider implementing a baseline pain threshold measurement in order to check whether this may have occurred.

## Pre-registration

In the pre-registration (see: https://aspredicted.org/k2vb9.pdf), increase in pain tolerance was presented as the main hypothesis and subjective pain as part of additional analyses, although it was made prior to conducting the study and later amended on the basis of the assumption that pain tolerance and subjective pain (i.e., pain intensity) would go hand-in-hand. Fortunately, the study design remained set up in such a way that allowed for both indices to be tested individually. Unfortunately, data pertaining to pain tolerance could not be analyzed as planned due to technical problems with the pain tolerance timer. The analyses were therefore restricted to the subjective pain ratings.

Additionally, it is important to note that the pre-registered exclusion criteria set forth for the purpose of controlling the pain tolerance measure (i.e., not complying with instructions to turn timer on/off) were not used because these did not interfere with the measure of subjective pain. Furthermore, participants in the SEX condition who reported low levels of sexual arousal

were not excluded because any threshold for exclusion would be arbitrary and could bias the results given that characteristics associated with low arousal reports may also be associated with participant pain ratings.

## Conclusion

The present findings do not support the hypothesis that sexual arousal alone modulates subjective pain in women. This might potentially be due to the possibility that genital stimulation and/or orgasm (instead of sexual arousal per se) are key in sex-related pain reduction, or, that feelings of disgust may inadvertently have been induced by the pornographic stimulus and interfered with sexual arousal in influencing pain intensity.

## Acknowledgments

Special thank you to Bert Hoekzema, Pieter Zandbergen, & Remco Willemsen, all of whom have contributed to setting up the laboratory and experiment materials, and greatly facilitated the realization of this project. Thanks as well to Lisa Maeder, who has written her Bachelor thesis on this project, for meticulously ensuring throughout that all the set up was in check in the laboratory for the experiment to run smoothly, as well as for supervising intern students.

## Author Contributions

**Conceptualization:** Lara Lakhsassi, Charmaine Borg, Sophie Martusewicz, Karen van der Ploeg, Peter J. de Jong.

**Data curation:** Lara Lakhsassi, Sophie Martusewicz.

**Formal analysis:** Lara Lakhsassi, Charmaine Borg, Sophie Martusewicz, Peter J. de Jong.

**Investigation:** Lara Lakhsassi, Charmaine Borg, Sophie Martusewicz, Peter J. de Jong.

**Methodology:** Lara Lakhsassi, Charmaine Borg, Sophie Martusewicz, Peter J. de Jong.

**Project administration:** Charmaine Borg, Peter J. de Jong.

**Supervision:** Charmaine Borg, Peter J. de Jong.

**Validation:** Charmaine Borg, Peter J. de Jong.

**Writing – original draft:** Lara Lakhsassi, Charmaine Borg, Peter J. de Jong.

**Writing – review & editing:** Lara Lakhsassi, Charmaine Borg, Sophie Martusewicz, Karen van der Ploeg, Peter J. de Jong.

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
