## [Decision Letter · Decision Letter 0]

14 Apr 2022

PONE-D-22-05538The influence of sexual arousal on subjective pain intensity during a cold pressor test in womenPLOS ONE

Dear Dr. Lakhsassi,

Thank you for submitting your manuscript to PLOS ONE. I was able to get advice from one expert reviewer. As you will notice, the reviewer was positive about your study, but raised several issues. In particular, it would be nice if you could share your data in a public repository. Also, the discussion reads more like a summary of the results, rather than an attempt to integrate current findings with those from other studies. If you feel that you can address all the points raised by the reviewer, we invite you to submit a revised version of the manuscript.

We look forward to receiving your revised manuscript.

Kind regards,

José A Hinojosa, Ph.D.

Academic Editor

PLOS ONE

Journal Requirements:

Reviewers' comments:

Reviewer's Responses to Questions

**Comments to the Author**

1. Is the manuscript technically sound, and do the data support the conclusions?

Reviewer #1: Partly

2. Has the statistical analysis been performed appropriately and rigorously? 

Reviewer #1: Yes

3. Have the authors made all data underlying the findings in their manuscript fully available?

Reviewer #1: No

4. Is the manuscript presented in an intelligible fashion and written in standard English?

Reviewer #1: Yes

5. Review Comments to the Author

Reviewer #1: The paper entitled "The influence of sexual arousal on subjective pain intensity during a cold pressor test in women" presents very interesting research of how sexual arousal may affect pain perception. Their results do not find an effect of perceived pain modulation on the use of sexual stimuli in young women. Overall, the paper is well written, and ideas are clearly exposed. In my opinion, this paper would be appropriate for the readership of this journal. However, there are several points that the authors will have to solve before this work could be published in "Plos One". Below are different comments in no particular order of importance.

General comments:

- It would be convenient for the authors to share their database during the review process, beyond that, once published, they can share their data.

- It would be useful to edit the figures to make them easier to understand.

- To facilitate the review process, it would be convenient to include line numbers on the manuscript pages.

Introduction:

- In general, the introduction gives a coherent review of the state of the art, gathering sufficient literature on the subject.

- An "Introduction" heading at the beginning of the section is necessary.

- Specific hypotheses on the expected effects should be included in the final part.

Method:

- The method presents the information scattered throughout the section. My recommendation is to make fewer sub-sections and to integrate information about the experiment, as well as the measurements and stimuli used, in the "Procedure" section.

- Were differences in pain thresholds between participants assessed in any way? This would be important to do as the pain experience is highly subjective and controlling and then manipulating it is one of the key elements when working with pain. If different groups per se have different pain thresholds, this could influence subsequent outcomes. If this was not controlled for, it would be necessary to justify why.

- Was the choice of the videos used based on any kind of pre-assessment or was it randomly selected?

- Sexual orientation and age characteristics are assessed with questionnaires, but these data are not reflected in any section afterwards. A table with this information could be included at the beginning of the results.

- Only one measure of pain was taken during the video?. It would be more convenient to take several pain measurements to generate the threshold. As I mentioned before, there are many variables that affect the perception of pain, and several measurements could reduce this influence.

- The section "Data Analyses Plan" should be renamed as "Data analyses" or something similar. In this section there are again too many sub-sections with fragmented information. My recommendation, again, is to integrate the information to make it easier to follow. It would also be useful to integrate the information on the software used into the text.

- The subsection "Hypothesis Testing" could be deleted or at least integrated into the previous ones.

- The abbreviation CPT is used for the first time in this section but has not been previously defined.

Results:

- The use of the same terms to refer to dependent variables and conditions is confused. This makes the reading of the results confusing and makes it difficult to follow the results. I advise using different names for both.

- Effect sizes are only reported for some analyses, it would be interesting to report them for all analyses.

- In general, the description of the results is complex to follow, perhaps because of what I said before about the naming of variables. I would revise the wording to make it clearer.

- In the last section of the results, an analysis is made excluding certain subjects. In my opinion the exclusion of these subjects is arbitrary, although the authors explain that it is a technique comparable to that based on mean scores. I believe that an exclusion strategy based on outlier analysis would be more convenient and could shed more light on the data.

Discussion:

- The discussion is sparse and does not allow for a detailed explanation of the main results. Much space is devoted to commenting on the results, but there is no explanation or comparison with other articles of the most relevant results, i.e., the null effect of sexual arousal on pain perception. In the limitations, some data are given on why sexual arousal may be perceived differently in women, but this information should be discussed in more depth in the corresponding part of the discussion.

- The first two paragraphs of the discussion could be integrated into a single paragraph in which only a summary of the results found is presented, followed by a paragraph discussing the most relevant results.

- In general, there is little bibliography in this section (there are only 6 references in the whole discussion). A more extensive literature search is needed to discuss the results found.

6. PLOS authors have the option to publish the peer review history of their article (what does this mean?). If published, this will include your full peer review and any attached files.

Reviewer #1: No

---

## [Author Response · Author response to Decision Letter 0]

12 May 2022

Dear Dr. José A Hinojosa and Expert reviewer, 

Thank you for the time you have dedicated to our manuscript, and for inviting us to revise our manuscript. We have submitted the requested documents and addressed all the points raised by the reviewer. In response to your comments, we have also submitted our data in DataverseNL; this is currently under review for anonymity and will be made available as soon as possible. We have furthermore attempted to integrate our findings with those from other studies to make the discussion section more elaborate, as suggested. 

We look forward to hearing your thoughts on the revised manuscript. 

Kind regards, 

Lara Lakhsassi, on behalf of all co-authors

---

## [Decision Letter · Decision Letter 1]

26 Aug 2022

The influence of sexual arousal on subjective pain intensity during a cold pressor test in women

PONE-D-22-05538R1

Dear Dr. Lakhsassi,

We’re pleased to inform you that your manuscript has been judged scientifically suitable for publication and will be formally accepted for publication once it meets all outstanding technical requirements.

Kind regards,

José A Hinojosa, Ph.D.

Academic Editor

PLOS ONE

Additional Editor Comments (optional):

Reviewers' comments:

Reviewer's Responses to Questions

**Comments to the Author**

1. If the authors have adequately addressed your comments raised in a previous round of review and you feel that this manuscript is now acceptable for publication, you may indicate that here to bypass the “Comments to the Author” section, enter your conflict of interest statement in the “Confidential to Editor” section, and submit your "Accept" recommendation.

Reviewer #1: All comments have been addressed

2. Is the manuscript technically sound, and do the data support the conclusions?

Reviewer #1: Yes

3. Has the statistical analysis been performed appropriately and rigorously? 

Reviewer #1: Yes

4. Have the authors made all data underlying the findings in their manuscript fully available?

Reviewer #1: Yes

5. Is the manuscript presented in an intelligible fashion and written in standard English?

Reviewer #1: Yes

6. Review Comments to the Author

Reviewer #1: This new version of the manuscript entitled "The influence of sexual arousal on subjective pain intensity during a cold pressor test in women" presents a much more comprehensible text. This new version of the manuscript submitted by the authors allows a much better follow-up of the study. They have also been able to respond to all the comments proposed by this reviewer. For all these reasons, in my opinion, this manuscript, in its current version, can be accepted by Plos One.

7. PLOS authors have the option to publish the peer review history of their article (what does this mean?). If published, this will include your full peer review and any attached files.

Reviewer #1: No

---

## [Editor Report · Acceptance letter]

26 Sep 2022

PONE-D-22-05538R1 

The influence of sexual arousal on subjective pain intensity during a cold pressor test in women 

Dear Dr. Lakhsassi:

I'm pleased to inform you that your manuscript has been deemed suitable for publication in PLOS ONE. Congratulations! Your manuscript is now with our production department. 

Kind regards, 

on behalf of

Dr. José A Hinojosa 

Academic Editor

PLOS ONE